# In vitro 3D drug sensitivity testing for patient-derived tumor-like cell clusters and whole-exome sequencing to personalize postoperative treatment: A study protocol for a multicenter randomized controlled trial

Xiaoyuan Qiu[1]ᵒ, Ruiqiang Li[2]ᵒ, Gengchen Xie[3], Ning Ning[4], Zhenjun Wang[5], Jiaolin Zhou[1], Ge Liu[6], Qingchao Tang[7], Zhuo He[8], Leilei Yang[2], Junyang Lu[1], Peng Li[9]*, Guole Lin[1]*

1 Department of General Surgery, Peking Union Medical College Hospital, Chinese Academy of Medical Sciences and Peking Union Medical College, Beijing, China, 2 GeneX Health Company Limited Laboratory, Beijing, China, 3 Department of Gastrointestinal Surgery, Union Hospital, Tongji Medical College, Huazhong University of Science and Technology, Wuhan, China, 4 Department of Gastrointestinal Surgery, Peking University International Hospital, Beijing, China, 5 Department of General Surgery, Beijing Chaoyang Hospital, Capital Medical University, Beijing, China, 6 Department of General Surgery, The First Affiliated Hospital, Dalian Medical University, Dalian, China, 7 Department of Center of Cancer, Colorectal Oncological Surgery, The Second Affiliated Hospital of Harbin Medical University, Harbin, China, 8 Department of Gastroduodenal and Pancreatic Surgery, Hunan Cancer Hospital, Affiliated Cancer Hospital of Xiangya Medical School, Central South University, Changsha, China, 9 Department of General Surgery, First Medical Center of Chinese People's Liberation Army General Hospital, Beijing, China

ᵒ These authors contributed equally to this work.
* linguole@126.com (GL); doctorlipeng@126.com (PL)

## Abstract

### Background

Colorectal cancer (CRC) ranks as the third most common cancer globally, with significant postoperative recurrence and metastasis rates. The heterogeneity of CRC presents challenges in the selection of adjuvant chemotherapy regimens, highlighting the need for personalized treatment strategies. The development of in vitro models for drug sensitivity testing, including a novel patient-derived tumor-like cell cluster (PTC) model, offers a potential solution for predicting drug efficacy and guiding treatment.

### Methods and design

This multicenter randomized controlled trial (RCT) aims to evaluate the consistency between the in vitro PTC drug sensitivity test results with whole exome sequencing and the clinical prognosis of CRC patients. The study will involve 200 patients who will be randomly assigned to receive either PTC-guided adjuvant chemotherapy or traditional chemotherapy. The primary endpoint is the 3-year disease-free survival rate (3yDFS), with secondary endpoints including the consistency between test

**Data availability statement:** No datasets were generated or analysed during the current study. All relevant data from this study will be made available upon study completion.

**Funding:** Guole Lin National High Level Hospital Clinical Research Funding (No. 2022-PUMCH-B-005).

**Competing interests:** The authors have declared that no competing interests exist.

results and clinical outcomes and the prognostic value of gene mutations and other biomarkers.

## Discussion

This study represents a significant step toward precision medicine in CRC treatment by integrating PTC technology with whole-exome sequencing. These findings could provide valuable insights into personalized treatment approaches, potentially improving the clinical outcomes of patients with CRC.

## Trial registration

ClinicalTrials.gov: NCT05424692. Registered on June 21, 2022.

---

## 1 Background

Colorectal cancer (CRC) is a common gastrointestinal malignancy that seriously endangers human health. According to cancer statistics 2023 [1], CRC has become the third most common cancer in terms of both incidence and mortality rates. Approximately 25–30% of stage II CRC patients experience recurrence and metastasis within 5 years after surgery, whereas approximately 50–60% of stage III CRC patients experience recurrence and metastasis within 5 years after surgery [2,3]. In addition, 45–50% of patients with stage II-III disease develop liver metastases within two years after primary resection [4,5]. As the main adjuvant treatment for CRC surgery, multiple international clinical studies have shown that surgery combined with adjuvant chemotherapy can achieve a total 5-year survival rate of 60–70% and a total 5-year disease-free survival rate of 55–65% for high-risk stage II and stage III patients [6,7].

However, CRC is a malignant tumor with significant heterogeneity [8,9], and there are significant individual differences in drug sensitivity among patients. Even patients with the same pathological type and clinical stage who receive the same treatment plan have different prognoses and outcomes [10]. The selection of adjuvant chemotherapy regimens for different subgroups of patients remains controversial. Therefore, constructing a model that can accurately predict drug efficacy and guide personalized treatment for patients is one of the strategies for achieving precision treatment of CRC. Scientists have been dedicated to developing in vitro models for drug sensitivity testing over the past century, including cell lines, primary tumor cells [11], patient-derived organoids (PDOs) [12], and patient-derived tumor xenograft (PDX) mice [13]. However, owing to issues such as standardization, accuracy, and cycle, these methods have certain limitations in clinical application and struggle to achieve true clinical translation.

In 2020, Professor Jianzhong Xi's team developed a novel in vitro model called patient-derived tumor-like cell clusters (PTCs) [14]. Three-dimensional microspheres are formed by the self-assembly of tumor cells and tumor stromal cells from patient samples after cutting, digestion, and proliferation. The resulting PTCs include various

cells derived from tumor tissue, such as tumor epithelial cells, tumor stem cells, fibroblasts, and immune cells, which can simulate the tumor microenvironment to a certain extent. The PTC model is highly similar to the original tumor tissue in terms of morphology, pathological characteristics, genetic background, and other dimensions [15]. For gastrointestinal tumors, the consistency between PTC drug sensitivity test results and clinical efficacy reached 96.6% [14]. Therefore, the high accuracy of PTC drug sensitivity testing technology has been preliminarily confirmed.

On the basis of the above background, this study aims to conduct a multicenter randomized controlled trial to compare the consistency between the in vitro PTC drug sensitivity test results and the clinical prognosis of CRC patients. The goal is to explore the decision-making value and guiding significance of this technology for assisting in precision treatment of CRC. This study will provide a valuable reference for achieving individualized and precise treatment of CRC patients and offer real-world data support for the clinical application of PTC drug sensitivity testing technology.

## 2 Methods and design

### 2.1 Study overview

This multicenter randomized controlled trial (RCT) will be conducted at 9 tertiary hospitals: Peking Union Medical College Hospital, Tianjin People's Hospital, Chinese PLA General Hospital, Beijing Chaoyang Hospital Affiliated with Capital Medical University, Huazhong University of Science and Technology Tongji Medical College Affiliated Union Hospital, Dalian Medical University Affiliated First Hospital, The Second Affiliated Hospital of Harbin Medical University, Cancer Hospital of Xiangya Medical College Central South University, and Peking University International Hospital. The study was designed following the Standard Protocol Items: Recommendations for Interventional Trials (SPIRIT) statement [16] (Fig 1). Participants who meet the inclusion criteria will be randomly assigned at a 1:1 ratio to either the PTC-guided adjuvant chemotherapy group (experimental group) or the traditional chemotherapy group (control group). These participants will then receive different interventions and undergo evaluations throughout the entire duration of the trial (see Fig 2 for a depiction of the study flow). Baseline data will be collected from participants at the beginning of the trial, followed by follow-up assessments at specific intervals. The primary endpoint of the study is the 3-year disease-free survival rate (3yDFS). Through this study, we aim to evaluate the results of PTC drug sensitivity testing in comparison with clinical outcomes, assess the consistency between the results of this technological platform and clinical prognosis, and provide a valuable reference for achieving individualized and precise treatment for CRC patients, thereby improving clinical benefits.

### 2.2 Participants and recruitment

We will recruit a total of 200 patients with clinically staged T3-4N0M0 or TanyN+M0 CRC from 9 hospitals in China. The study will document the demographic data, clinical and pathological features, imaging findings, operative results, and oncological outcomes of all enrolled patients.

At this point, we have included all patients clinically staged as T3-4N0M0 or TanyN+M0 in the candidate pool and obtained signed consent forms for the study, but the patients have not yet been enrolled. If patients are determined to require radical resection for CRC after clinical diagnosis and treatment, we will collect approximately 50 mg of fresh cancer tissue with abundant blood vessels from the excised tumor specimens in the operating room after the completion of radical surgery. The samples will be placed in preservation tubes and transported to the laboratory at 4 °C, where PTC model construction will be completed. Upon receipt of the surgical pathology results, if patients indeed meet the criteria for adjuvant chemotherapy, we will select patients on the basis of the inclusion and exclusion criteria.

**2.2.1 Inclusion criteria.** The inclusion criteria are as follows: 1) patients with histologically confirmed CRC; 2) baseline clinical stage of cT3-4N0M0 or cTanyN+M0; 3) CRC patients requiring adjuvant treatment after radical surgery and not receiving neoadjuvant treatment; 4) at least one evaluable tumor lesion; and 5) Eastern Cooperative Oncology Group (ECOG) performance status ≤ 2.

| | STUDY PERIOD | | | | | | | | | | | | | | |
|---|---|---|---|---|---|---|---|---|---|---|---|---|---|---|---|
| | Enrolment | Allocation | Post-allocation | | | | | | | | | | | | Close-out |
| TIMEPOINT** | -14d | 0 | 3m | 6m | 9m | 12m | 15m | 18m | 21m | 24m | 30m | 36m | 48m | | 60m |
| **ENROLMENT:** | | | | | | | | | | | | | | | |
| Eligibility screen | X | | | | | | | | | | | | | | |
| Informed consent | X | | | | | | | | | | | | | | |
| PTC model construction | X | | | | | | | | | | | | | | |
| WES | X | | | | | | | | | | | | | | |
| Pathology results | | X | | | | | | | | | | | | | |
| Allocation | | X | | | | | | | | | | | | | |
| **INTERVENTIONS:** | | | | | | | | | | | | | | | |
| Adjuvant chemotherapy regimens on the basis of the PTC drug sensitivity test results | | | ——— | | | | | | | | | | | | |
| Adjuvant chemotherapy regimens on the basis of clinical experience | | | ——— | | | | | | | | | | | | |
| **ASSESSMENTS:** | | | | | | | | | | | | | | | |
| Baseline assessments | X | X | | | | | | | | | | | | | |
| PTC characteristic and WES results | X | X | | | | | | | | | | | | | |
| Follow-up assessments | | | X | X | X | X | X | X | X | X | X | X | X | | X |
| Survival status | | | X | X | X | X | X | X | X | X | X | X | X | | X |

**Fig 1. Schedule of enrollment, interventions and assessments.**

**2.2.2 Exclusion criteria.** The exclusion criteria are as follows: 1) patients with distant metastases; 2) patients unable to obtain tumor samples; 3) pregnant and lactating women; 4) patients with poor compliance; 5) patients with severe cardiovascular and cerebrovascular complications unable to receive chemotherapy or targeted therapy; 6) patients previously diagnosed with other malignant tumors; 7) patients with severe mental and neurological disorders; and 8) patients deemed unsuitable for participation in this study by researchers.

**2.2.3 Participant withdrawal.** Participants have the right to withdraw from the study at any time without providing a reason. Any participants who are lost to follow-up will be recorded, and an intention-to-treat analysis will be performed exclusively for these individuals.

### 2.3 Blinding and randomization

Considering that the subjects in our study will be aware of what kind of chemotherapy they receive, we will not utilize the double-blind method [17]. To reduce potential bias and ensure fair allocation of participants, we designed blocks of size

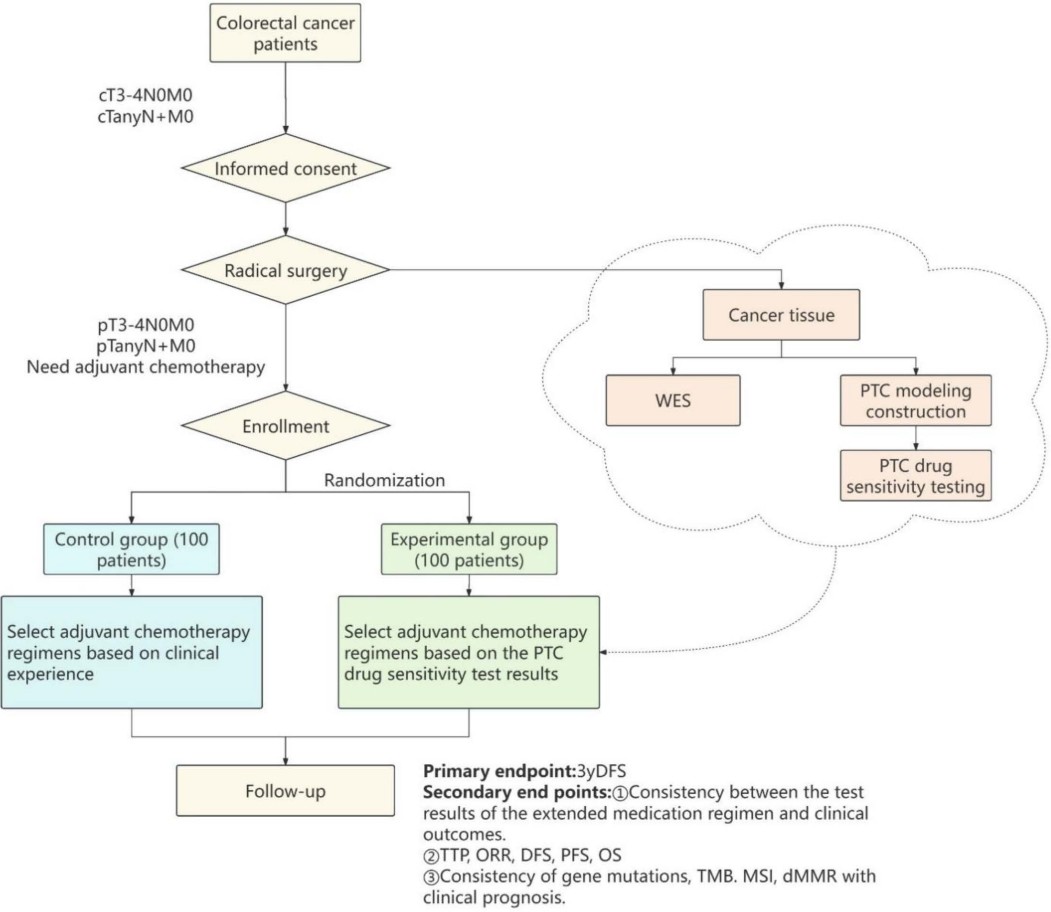

**Fig 2. Flow chart of the study.**

four and randomly assigned patients to either the PTC adjuvant chemotherapy group or the traditional chemotherapy group at a 1:1 ratio. Stratification was performed according to the surgical center [18].

## 2.4 Intervention

We plan to include 200 patients for analysis, with 100 in the experimental group and 100 in the control group. Both groups will undergo PTC drug sensitivity testing and whole exome sequencing (WES). For the experimental group, doctors will select adjuvant chemotherapy regimens on the basis of the PTC drug sensitivity test results. For the control group, doctors will determine adjuvant chemotherapy strategies on the basis of their clinical experience. The PTC drug sensitivity and WES results will be documented for humanitarian treatment reference after the study concludes or if patients withdraw from the study.

## 2.5 Therapeutic schemes

**2.5.1 PTC drug sensitivity test.** Upon receiving the tumor tissue, the laboratory will verify the sample number and establish a sample file via the laboratory management system for data file management. After laboratory tests are conducted on the sample status (including temperature, time in vitro, and presence of leakage), sample processing and

cultivation will be performed. After 2–7 days of cultivation, the formation of PTC clones will be observed via bright field microscopy. If PTC clones fail to form within 7 days, cultivation will be declared unsuccessful, and the patient will be withdrawn. If PTC clones are formed, open field photos will be retained to check for bacterial, fungal, and mycoplasma infections in the culture. Samples identified as negative will be subjected to drug sensitivity experiments [14].

We will test four commonly used adjuvant chemotherapy regimens in clinical practice [19]: fluorouracil+ formyltetrahydrofolate (5-RU) [7], oxaliplatin+5-fluorouracil+formyltetrahydrofolate (FOLFOX) [20], irinotecan+ 5-fluorouracil+formyltetrahydrofolate (FOLFORI) [21,22], and cetuximab+5-fluorouracil+formyltetrahydrofolate [23,24]. For the screening experiment, a 96-well plate will be used, and each experimental group will be equipped with three parallel wells to ensure the reliability of the experimental data. Each sample will be collected twice, once after 0 days of dosing and once after 7 days of dosing. The automatic scanning system will be used to identify the number and area of all PTC clones in the hole. The area ratio of PTC clones (i.e., residual cell viability) after and before dosing will be calculated via the $p_{Ai} = \frac{S_{Ai,t}}{S_{Ai,t0}}$, $p_A = \frac{1}{n}\sum_{i=1}^{n} p_{Ai}$ formula [25], and the interpretation criteria are shown in Table 1. The higher the score is, the better the drug's efficacy in killing tumors, and higher-scoring regimens are more recommended.

**2.5.2 Whole-exome gene sequencing (WES).** We will perform whole-exome gene testing on the patient's histological specimens. The specific testing procedure is as follows: 1) tumor-targeted genes, including point mutation, insertion/ deletion, fusion, and copy number variation analyses, will be used to predict the effectiveness of targeted drugs, tumor molecular subtypes, and patient prognosis; 2) a total of 145 tumor chemotherapy-related genes will be used to predict the effectiveness and side effects of commonly used chemotherapy drugs; 3) tumor mutational burden and microsatellite instability will be used to predict the efficacy of immunotherapy drugs; and 4) tumor susceptibility genes will be used to predict the genetic risk of multiple tumors.

## 2.6 Follow-up and endpoints

The patient's age, sex, pathological type, treatment plan, treatment drug dosage and cycle, pretreatment clinical staging, imaging results, oncological outcomes, etc., will be recorded. After discharge, patients will be reexamined in the outpatient clinic two to three weeks postsurgery to record any short-term complications and will subsequently be followed up for five years. Within the first two years postsurgery, patients will undergo clinical examinations every three months, including a digital rectal examination (DRE), examination of tumor markers such as CEA and CA19−9, chest X-ray (front and side views), and abdominal ultrasound (liver, gallbladder, pancreas, spleen, and kidney). Within two to three years postsurgery, patients will be followed up every six months. After three years, annual follow-ups will be conducted, which will include chest–abdomen–pelvis enhanced CT scans, rectal MRI or rectal ultrasound, and colonoscopy.

Follow-up will be conducted through doctor outpatient reviews as well as specialized follow-up personnel via phone, e-mail, text and other means. During the follow-up process, detailed records of the patient's status (tumor-free survival, local recurrence, distant metastasis, death, etc.) will be recorded. The time, location, and cause of the endpoint event will also be recorded.

Table 1. Criteria for Judging the Drug Sensitivity of PTCs.

| Lethality level | Lethality cell percent (%) | Remaining cell percent (%) |
| --- | --- | --- |
| Strong killing effect | >0.7 | 0-0.3 |
| Effective killing effect | 0.3-0.7 | 0.3-0.7 |
| Stable | 0.1-0.3 | 0.7-0.9 |
| Drug resistant | 0-0.1 | 0.9-1 |
| Strongly drug resistant | 0 | 1 |

The main endpoint of the study is the 3yDFS rate. The secondary endpoints of the study are as follows: 1) consistency between the test results of the extended medication regimen and the clinical outcomes; 2) time to progression (TTP), objective response rate (ORR), progression-free survival (PFS), and overall survival (OS); and 3) consistency of gene mutation status, tumor mutation burden (TMB), microsatellite instability (MSI), and mismatch repair (MMR) status with clinical prognosis.

## 2.7 Statistical analysis

**2.7.1 Sample size.** Our study is a noninferiority study. With a bilateral $\alpha = 0.05$, a test efficacy of $1-\beta = 80\%$, a 3-year disease-free survival rate of 75%, and a noninferiority threshold of 20%, the estimated minimum sample size for each group is 74. Considering a dropout rate of 20% (according to EMEA [26], ICH [27], and FDA [28] guidelines and previous studies [29]), we ultimately decided to include 100 patients in each group, totaling 200 patients across both groups.

**2.7.2 Data collection.** Researchers are required to fill in the collected data on paper case report forms according to the research protocol requirements, and trained personnel will input them into the specially established electronic data capture (EDC) system. At the end of the study, researchers will submit case report forms for all selected patients to the data management center, which should include complete and signed content. Personally identifiable participant information, such as name and gender, will be replaced by codes throughout the study and will remain confidential, with only the project manager or relevant doctors aware of the patient's personal information. Any reports on the results of this study will not reveal any identifiable information about the patients.

**2.7.3 Statistical methods.** Local recurrence is defined as tumor recurrence in the lumen or in the mesorectum and must be confirmed by a pathological biopsy. A distant metastasis is defined as a metastasis that is discovered in the liver, lung, bone, or other sites by CT, MRI, or radionuclide scanning, with or without pathological examination. Disease-free survival (DFS) is defined as the absence of local recurrence, distant metastasis, and death from any cause. Overall survival (OS) is defined as the absence of death from any cause. The duration of follow-up will be calculated from the date of surgery to the event of interest or the last follow-up date.

To determine the consistency between the PTC drug sensitivity results and clinical outcomes, the following methods will be employed (Table 2): positive consistency rate $= a/(a+c) \times 100\%$; negative consistency rate $= d/(b+d) \times 100\%$; overall consistency rate $= (a+d)/(T1) \times 100\%$; sensitivity $= a/(a+b) \times 100\%$; and specificity $= d/(b+d) \times 100\%$.

The results will be presented as median values or means ± standard deviations for continuous variables and as numbers (percentages) for categorical variables. The t test and Wilcoxon rank sum test will be used to compare continuous variables, and chi-square analysis/Fisher's exact test will be used to compare categorical variables. Survival analysis will be conducted using the Kaplan-Meier curves. Log-rank test to compare the primary endpoint between different arms in the presence of censoring. A P value of 0.05 or less is considered statistically significant. All the statistical tests will be two-sided. All analyses will be performed using SPSS Version 26.0 (SPSS Inc., Chicago, IL).

## 2.8 Quality control

To ensure that the quality of research activities meets the requirements, monitoring will be carried out at each stage of data processing to ensure that all the data are trustworthy and processed correctly. The sponsor is responsible

**Table 2. Consistency test of recurrence and metastasis cases.**

| Drug sensitivity test | Clinical efficacy | | Sum |
|---|---|---|---|
| | CR+PR | SD+PD | |
| Effective | a | B | a+b |
| Invalid | c | D | c+d |
| Sum | a+c | b+d | T1 |

for comprehensive tracking and monitoring of the implementation of clinical trials, ensuring that the experiments comply with relevant regulatory requirements and follow the experimental protocol. Clinical trial researchers are familiar with necessary regulatory requirements, possess professional expertise related to clinical trials, have the ability to design and implement relevant clinical trials, are familiar with relevant testing techniques, can correctly interpret test results, have passed qualification review, and are relatively fixed personnel. Laboratory quality control includes establishing unified laboratory standard operating procedures and quality control procedures and conducting quality control simultaneously during drug sensitivity testing. Only when the quality control results are qualified can the data be determined to be valid. The person in charge of statistics has a relevant professional background and abilities.

A double data check will also be performed for quality control. In accordance with the data collection methods and criteria developed by the project leader and professionals, all the data will be recorded carefully and comprehensively to ensure the authenticity of the data. Missing data from withdrawn participants will be imputed via the multiple imputation method.

## 2.9 Ethics and dissemination

All the subjects will participate in the study voluntarily, and the senior researchers on our project team, Dr. Guole Lin, Dr. Peng Li, Dr. Ning Ning, Dr. Zhenjun Wang, Dr. Gengchen Xie, Dr. Ge Liu, Dr. Qingchao Tang, and Dr. Zhuo He, will obtain written consent forms from each individual prior to their enrollment in the study. Both groups of subjects are expected to benefit from the study. Additionally, we will inform all the participants about any new information that may influence their willingness to participate in the study. The trial adheres to the criteria and principles outlined in the Declaration of Helsinki and received approval from the Ethics Committee of Peking Union Medical College Hospital (project ID: HS-3174B). This study will be subject to an annual ethical review. Furthermore, we registered the trial on chictr.org.cn. Any modifications to the study will be promptly communicated to the Ethics Committee of Peking Union Medical College Hospital and will be updated on the registration website once approval is obtained.

As the Chinese Clinical Trial Registry still allowed delayed registration after study began on 2021, we initiated recruitment after obtaining ethical approval. Sometime later, we realized that ClinicalTrials encourages authors to register before recruitment. Thus, we conducted a supplementary registration immedatly, which led to the trial being registered after patient recruitment had begun. We understand the importance of timely registration for the transparency, ethics, and credibility of research results, and we also recognize the potential impact of this delay. The authors confirm that all ongoing and related trials for this drug/intervention are registered.The entirety of the research-related data, including original data and records, quality control files, and software used for data storage and follow-up analysis, will be limited to access by the study team only. The statistical analysis and processing of the collected data will be conducted exclusively by designated researchers. The identity and contact information of all participants will be strictly confidential. Prior to performing the statistical analysis, the data will be anonymized. Ultimately, the research findings will be published in reputable scientific journals.

## 3 Discussion

Colorectal cancer, a highly prevalent malignant tumor, has a significant impact on patients' survival. To improve patient survival, for all suitable stage III colon cancer patients, as well as stage II colon cancer patients with high-risk features (T4 tumor, tumor perforation, fewer than 12 resected lymph nodes), adjuvant (postoperative) chemotherapy is recommended after curative tumor resection [30]. Adjuvant chemotherapy should be considered for all clinical and/or histological stage II and III rectal cancer patients [31]. Drugs such as oxaliplatin and fluoropyrimidine play crucial roles in the treatment of locally advanced and metastatic CRC as first-line therapies, enhancing local tumor control and improving survival outcomes.

However, an increasingly prominent issue is the development of primary or acquired resistance to these drugs in a considerable proportion of patients [32,33], which is a major factor affecting the prognosis of CRC patients [34]. Up to 40% of patients who receive 5-FU-based adjuvant chemotherapy after resection for stage II and III colon cancer still experience recurrence or death within 8 years of follow-up [35]. Intratumoral heterogeneity is an inherent factor responsible for chemotherapy resistance [36], resulting from various genetic, epigenetic, transcriptomic, and proteomic characteristics of tumor cells [37]. Therefore, determining patients' sensitivity to chemotherapy drugs and guiding personalized treatment for patients is highly important for clinicians. Scientists have been dedicated to developing in vitro drug sensitivity testing systems over the past century to assess tumor responses to different drugs.

In 1988, Sevin et al. pioneered the application of adenosine triphosphate biofluorescence tumor drug sensitivity detection technology (ATP-TCA) in 2D cultured ovarian cancer [11], but 2D models are incapable of replicating the spatial structure and heterogeneity of tumors within the body, nor can they capture the interactions between tumor cells and the tumor microenvironment (TME). Subsequent research focused on optimizing cultivation methods for 3D models, which better mimic in vivo conditions. For example, tumor spheroids, which better retain the morphological, physicochemical, and genetic characteristics of solid tumor cells, play crucial roles in the study of CRC invasion, metastasis, drug resistance mechanisms, and the development of targeted therapies and novel immunotherapies [38,39]. However, this technique faces challenges such as complex culture methods, a lack of vascular systems, disparities from real conditions, and difficulty in long-term stable cultivation.

PDOs are the result of a 3D cultivation technique that has gained attention in recent years. In 2015, M. van de Wetering et al. first reported the successful use of PDOs for high-throughput drug screening in CRC [40]. In 2018, G. Vlachogiannis et al. confirmed that PDOs cultured with tumor stem cells accurately predict the efficacy of anticancer drugs [12]. PDOs demonstrated 100% sensitivity and 93% specificity, with an 88% positive predictive value and 100% negative predictive value. Compared with spheroids, tumor organoids are highly complex and closely resemble the cellular composition, morphological structure, and physicochemical and genetic characteristics of tumors in vivo; they are also more suitable for biological transfection and can be cultured stably for longer periods in vitro. This makes organoids useful for high-throughput drug sensitivity studies, personalized precision therapy, and the establishment of biobanks. However, there are limitations. The success rate of cultivating CRC organoids is not yet satisfactory, particularly for metastatic CRC cases where cancer tissue can be obtained only through biopsy, with a modeling success rate of only approximately 70% [40,41].

In addition, researchers have transplanted tumor tissue into immunodeficient mice and established a PDX model in which mice are used instead of humans for drug sensitivity testing [13,42]. However, owing to limitations such as success rates (50% or even lower), testing cycles (more than 4 months), and high costs, it is difficult to apply this model in clinical practice.

As a novel in vitro tumor model, PTCs can form 3D microspheres by cutting, digesting, and proliferating samples from patients and then self-assembling tumor cells and tumor stromal cells, which are highly similar to the original tumor tissue in terms of morphology, pathological features, genetic background, and other dimensions [14]. This technology is an efficient, accurate, and short-cycle in vitro drug sensitivity detection system. PTC drug sensitivity detection technology overcomes the bottleneck issues of in vitro model technologies in clinical applications, achieving trace amounts, safety, accuracy, and timeliness, and has great potential in clinical translational applications.

For patients after tumor surgery, it is difficult to evaluate whether adjuvant chemotherapy drugs can effectively kill the tumor, as there is no longer any visible tumor on imaging. On the basis of PTC technology, this study will conduct drug sensitivity tests in vitro after surgery and before adjuvant therapy to evaluate the sensitivity of tumors to chemotherapy drugs to implement precise and personalized treatment for patients. Ultimately, our goal is to provide personalized adjuvant chemotherapy regimens for patients with colorectal cancer, thereby increasing survival benefits.

Previous studies on PTC technology have been cases reports, retrospective or small-scale single center observational studies. This study is the first multicenter, prospective randomized controlled trial to explore the sensitivity of adjuvant

chemotherapy regimens for CRC using PTC as a new technology. If this method appears beneficial, this study can help to generate preliminary guidelines for postoperative chemotherapy management in CRC patients. We will disseminate the results of this study to international journals and conferences.

## Time points/timeline, trial status, and publication plan

Recruitment for this RCT commenced in 2021, with the first patient enrolled on November 9, 2021. The recruitment was initially planned to be completed by 2024. However, due to the impact of the COVID-19 pandemic, recruitment for this study has been significantly delayed, and the enrollment deadline has been extended to September 1, 2025. Publication based on Protocol Version 1.0: 28 September 2021. At the time of publication, all sites are open to recruitment.

## Supporting information

**S1 File. Full text of protocol.**
(PDF)

**S2 File. SPIRIT Checklist.**
(DOC)

## Acknowledgments

We thank the personnel (research doctors, research nurses, and audiologists) from all the hospitals involved in this research.

## Author contributions

**Conceptualization:** Gengchen Xie, Ning Ning, Jiaolin Zhou, Leilei Yang, Peng Li.

**Data curation:** Xiaoyuan Qiu, Junyang Lu, Peng Li.

**Formal analysis:** Ruiqiang Li, Junyang Lu.

**Funding acquisition:** Guole Lin.

**Investigation:** Ruiqiang Li, Zhuo He.

**Methodology:** Ruiqiang Li, Jiaolin Zhou, Qingchao Tang, Leilei Yang, Peng Li, Guole Lin.

**Project administration:** Gengchen Xie, Ning Ning, Zhenjun Wang, Ge Liu, Zhuo He, Peng Li, Guole Lin.

**Resources:** Gengchen Xie, Ning Ning, Zhenjun Wang, Jiaolin Zhou, Ge Liu, Qingchao Tang, Zhuo He, Peng Li, Guole Lin.

**Supervision:** Gengchen Xie, Zhenjun Wang, Jiaolin Zhou, Ge Liu, Qingchao Tang, Peng Li, Guole Lin.

**Validation:** Ruiqiang Li, Ning Ning, Leilei Yang.

**Writing – original draft:** Xiaoyuan Qiu, Ruiqiang Li.

**Writing – review & editing:** Jiaolin Zhou, Peng Li, Guole Lin.

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
